# Hate and False Metaphors: Implications to Emerging E-Participation Environment

Sreejith Alathur [1,*], Naganna Chetty [2], Rajesh R. Pai [3], Vishal Kumar [4] and Sahraoui Dhelim [5,*]

1 Information Systems, Indian Institute of Management Kozhikode, Kozhikode 673570, India
2 A. J. Institute of Engineering and Technology, Mangaluru 575006, India
3 Department of Humanities and Management, Manipal Institute of Technology, Manipal Academy of Higher Education, Manipal 576104, India
4 Bipin Tripathi Kumaon Institute of Technology, Dwarahat 263653, India
5 School of Computer Science, University College Dublin, Belfield, D04 V1W8 Dublin, Ireland
* Correspondence: asreejith@iimk.ac.in (S.A.); sahraoui.dhelim@ucd.ie (S.D.)

**Abstract:** This study aims to investigate the effect of metaphorical content on e-participation in healthcare. With this objective, the study assesses the awareness and capability of e-participants to navigate through healthcare metaphors during their participation. Healthcare-related e-participation data were collected from the Twitter platform. Data analysis includes (i) awareness measurements by topic modelling and sentiment analysis and (ii) participation abilities by problem-based learning models. Findings show that a lack of effort to validate metaphors harms e-participation levels and awareness, resulting in a problematic health environment. Exploring metaphors in these intricate forums has the potential to enhance service delivery. Improving web service delivery requires valuable input from stakeholders on the application of metaphors in the health domain.

**Keywords:** capability awareness; e-participation; healthcare; India; metaphors; social media





## 1. Introduction

Internet technologies significantly helped to reduce the impact of coronavirus disease (COVID-19) by providing support for communications and economic activities [1,2]. Unlike during past pandemics that occurred at intervals, the government and citizens continued their operations in providing and availing themselves of services, including education and health, by using the internet [3,4]. Doctors were able to access different forums on the internet for the exchange of medical information [5]. However, the opportunities to avail themselves of such services and overcome the impact of the pandemic varied and this is not just because of the lack of access to technologies, but also because of the structure and evolution of internet-enabled communications.

Infectious disease is regarded as a global challenge, and social media health intervention has become a promising research domain. Irrespective of gender, infection attacks weaken the target's body. People with pre-existing health issues such as disability, obesity, and other chronic diseases are prone to infections more than others [6–8]. As vaccination is essential and still not all global citizens have been vaccinated, it is difficult to control the spread of infection. Therefore, social media has in recent times triggered the citizen protective actions that need to be practiced.

An understanding of complex incidents may not be feasible directly, but can be made easier through known information about other incidents. Metaphors are referred to as thinking and talking about something in terms of other things and drawing some similarities between dissimilar incidents [9]. Health-related and sensitive incidents such as illness can be deliberated on metaphorically [10]. There is a common practice of using metaphors when discussing a significant incident. As infections are a major disaster in the history of humanity, their deliberation involves a large number of metaphors. In

metaphorical statements, the use of terms such as enemy, tsunami, and glitter is prevalent against infections [9].

Capability and awareness are the two important factors for properly analyzing a situation. The effectiveness of e-participation means the ability of a citizen to perform e-participation using appropriate capabilities [11]. Lack of competence may lead to a wrong analysis of the situation. The absence of both competence and awareness often leads to a wrong intuition. Autonomy, competence, and relatedness are the basic requirements for the success of an individual and a group in society [12]. Self-regulating an individual required for well-being is possible through autonomy, competence, and relatedness [13].

In this digitized world, the expression of emotions on social media about a particular incident is prevalent. Emotion is a powerful but unconscious process that responds vigorously to problematic incidents [14]. As a result, it is established as a goal to assess the capacity of citizens who participate in discussions about infection to assist individuals with disabilities. The analysis of competence is made using emotions shared on social media involving disability during infections. The work also tries to identify the tweeting patterns of social media users. For these purposes, topic modelling and sentiment analysis are carried out using Twitter data.

The rest of the paper is structured as follows. Section 2 briefs the literature involving the research on infections, metaphors, emotions, competence, and unawareness. In Section 3, the methodology of the research is outlined. Results are presented in Section 4. Section 5 discusses the results. Finally, Section 6 concludes the work carried out.

## 2. Literature Review

Infectious disease has a multi-facet impact. The impact can be on individuals or a group. As the impact on an individual, the infection and associated responses may lead to emotional and psychological issues [15]. The impact on the group extends to organizations such as educational institutions, public and private sector units, and healthcare systems, among others. Educational institutions struggle to manage teaching and learning within the allotted time frame [16]. The hotel industry is also badly affected by the infection and its control strategies are equally affected by the governing authorities [17].

The emergence of infectious disease had led to persuasive expressions or metaphors by politicians, journalists, and other netizens [18]. To convince citizens of the repercussions of infectious disease, war metaphors are used. The prime ministers of different nations have used metaphors such as "a long battle ahead" to speak of disease [19]. On the other hand, anthropologists argue that infectious disease is not a war; analogizing it with war is wrong and may lead to dangerous incidents [20]. The use of war metaphors in medicine is inappropriate due to the different purposes of medicine and war, the fear that war instills in patients, and the availability of positive alternatives [21]. The use of war metaphors for infectious disease as a whole has both positive and negative effects.

Awareness in human beings develops gradually. There is no clear definition of awareness, but its usage is prevalent in different ways. Awareness refers to being responsive, awake, or knowing something [22]. Emotions play an important role in human activities [23] and provide awareness of objects [23,24]. Specifically, primordial emotions play a major role in the evolution of awareness [25]. There is an interconnection between an individual's emotions, self-awareness, and personality [26]. Self-awareness and the necessity of emotion influence an individual's attitude to femvertising [27].

The different kinds of awareness functions incorporating an individual's attitudes are extroverted and introverted sensations; extroverted and introverted intuitions; extroverted and introverted thinking; and extroverted and introverted feelings [28]. Emotions often trigger an individual's behaviour [29]. When an emotion initiates, the individual acts in a different way than normal. Emotions and cognitive functions are interconnected. The cognitive functions expand emotional intelligence [30]. The emotions of fear, anticipation, sadness, and disgust are associated with introverted cognitive functions and represent negativity in the individual. On the other hand, the emotions of anger, surprise, joy, and

trust are associated with extroverted functions and represent positivity in an individual [30]. The emotional response mediates the purchase and consumption of harmful items [31]. Control of emotions plays a significant role in shaping the behaviour of an individual for the well-being of society.

## 3. Materials and Methods

The methodology used in the study was a three-step process. First, the data were collected from Twitter social media by establishing a connection using an application programming interface (API) and authenticating with the Twitter developer identification information. The data gathering was carried out in the year 2021 and resulted in 483,880 tweets. The tweets were collected weekly and written in a separate file with a comma-separated values (CSV) format. After the data collection process, all files were merged to produce a single file containing all tweets.

In the second step of the methodology, the tweets were pre-processed. A tweet consists of sixteen attribute values. As we are interested in the opinions of the respondents, we considered only the textual part of the tweets for analysis. After extracting the text part of the tweets, the pre-processing steps were applied to eliminate punctuation, extra blank spaces, digits, and stop words from the tweets. As part of the pre-processing, the tweets were also tokenized and tokens were stemmed to the root of the token. At this point, the cleaned content of the tweets was ready for analysis.

In the third step, a function was applied to analyze the emotional score of the tweets as sentiments. This function resulted in different emotions and their score for the entire Twitter content. After the sentiment analysis, the word cloud function was used to identify the set of frequent words belonging to each emotion. After the word cloud analysis, topic modelling was performed to identify the co-occurrences of words. Finally, the results of all the analyses were interpreted and inferences were drawn collectively. All the pre-processing steps and functions for analysis were implemented in an open-source R programming language.

## 4. Results

After the data collection from Twitter social media, the program developed in R programming language was used to process and analyze the content. The entire dataset was divided into three parts: April 2021, May 2021, and June 2021, based on the months of tweets tweeted by social media users. The analysis was made using month-long datasets to identify the variations in social media users' tweeting patterns (capability and awareness) concerning infectious disease and disability.

Different emotions together with the frequency of occurrence in tweets are provided in Figure 1a. The analysis is made for eight basic and two sentimental emotions. 132.55K (132.55 thousand) tweets possess overall positive emotions and 81.86K tweets contain negative emotions. Most of the tweets (73.30K) contain basic emotional trust and depict the confidence of the social media users during the infectious disease. The next highest (59.29K) exhibited the emotion that fear is unpleasant and depicted a panic response to the incident. The anticipation emotion that hopes for the future was contained in 57.77K tweets. Sorrow was depicted through the emotion of sadness in 42.61K tweets.

There were 34.39K tweets containing the emotion of joy, which depicts pleasure. There were 30.17K tweets containing the necessary emotion of anger. There were 23,600 tweets depicting the emotional shock caused by the unexpected occurrence of an incident. The least-depicted emotion was disgust, which appeared in 18.44K tweets.

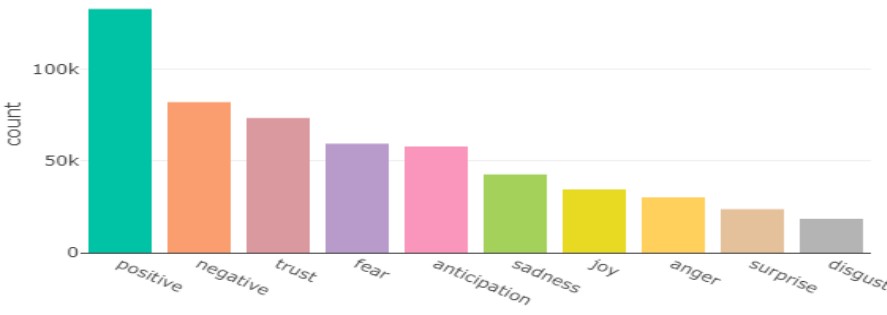

(**a**). Emotions during infectious disease in April 2021

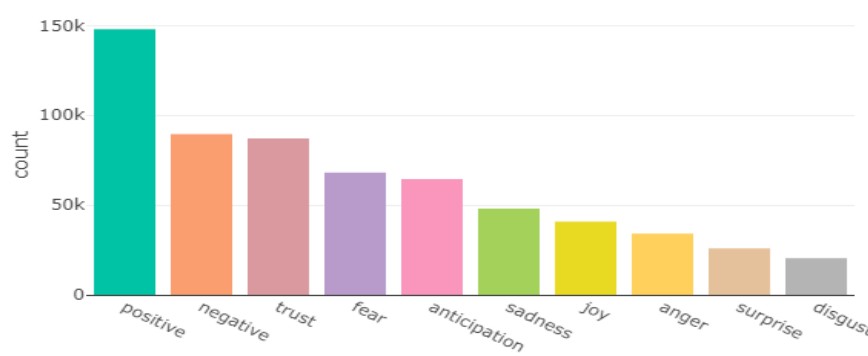

(**b**). Emotions during infectious disease in May 2021

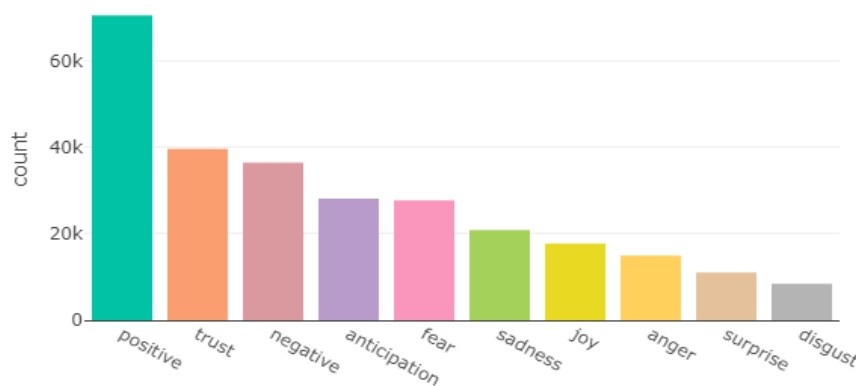

(**c**). Emotions during infectious disease in June 2021

**Figure 1.** Emotions during infectious disease.

The most frequent words concerning different emotions are shown in Figure 2. For brevity, only five frequent words are mentioned to represent each emotion. Emotion anticipation is mostly depicted through the frequent words time, public, tomorrow, long, and watch. Fight, strike, destruct, court, and demolish are frequent words used to show anger emotion. The emotion of trust is depicted through vaccine, mask, wear, nation, and team words. The frequent words good, highest, hope, urgent, and case show surprise emotion. The emotion of sadness is shown by using the words pandemic, hospital, blood, and plasma. Safe, stay, happiness, share and save are the words used to exhibit the emotion of joy. The emotion of fear is shown through words such as governance, risk, spike, infect and spread. Disgust is depicted through death, disease, sick, which are both negative and positive frequent words.

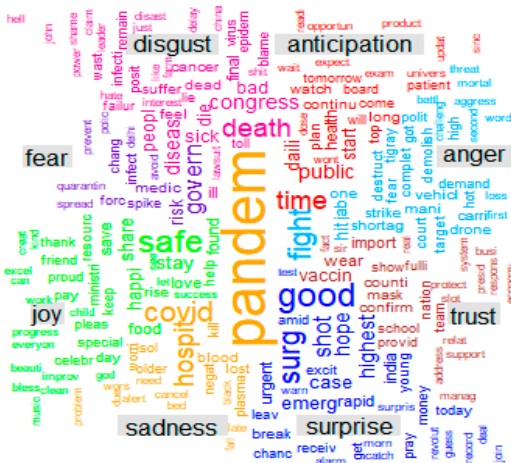

**Figure 2.** Wordcloud of emotions during infectious disease in April 2021.

Different emotions along with the frequency of occurrence in tweets of May 2021 are provided in Figure 1b. In this, 147.97K tweets contain overall positive emotions and 89.54K tweets contain negative emotions. Most of the tweets (87.08K) contain basic emotional trust and depict the confidence of the social media users during the infectious disease. The emotion of fear is exhibited through 68.10K tweets to depict a feeling of panic about the situation. The anticipation emotion is contained in 64.51K tweets. The sadness emotion occurred in 48.07K tweets to depict sorrow. The emotion of joy occurred in 40.87K tweets. Anger appeared in 34.18K tweets. Surprise was depicted in 25.97K tweets. The least depicted emotion, disgust, appeared in 20.43K tweets.

The most frequent words for emotions in May 2021 are shown in Figure 3. The emotion of anticipation is depicted through time, public, tomorrow, long, and watch as frequent words. Fight, jab, hit, court, and battle are frequent words used to show anger. Trust is depicted through words such as mask, import, oxygen, wear, and team. The frequent words good, hope, shot, urgent, and highest show the emotion of surprise. The emotion of sadness is shown through words such as pandemic, case, hospital, blood, and plasma. As in April 2021, safe, stay, happiness, share and save are used to exhibit the emotion of joy. The emotion of fear is shown through the words governance, quarantine, infect, and spread. The emotion of disgust is depicted through death, disease, fungus, suffering, and other frequent negative words.

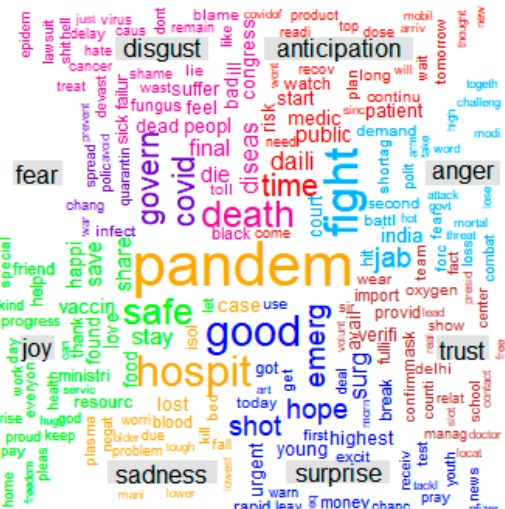

**Figure 3.** Wordcloud of emotions during infectious disease in May 2021.

Different emotions in June 2021 along with their frequency of occurrence are shown in Figure 1c. The overall positive and negative emotions in tweets are 70.53K and 36.50K, respectively. The basic emotion of trust is depicted in 39.64K tweets. The anticipation emotion is contained in 28.12K tweets. The emotion of fear is exhibited through 27.71K tweets to depict a sense of panic about the situation. The sadness emotion occurred in 20.85K tweets to depict sorrow. There were 17.73K tweets about the emotion of joy. Some 14.99K tweets contained the emotion of anger, while 11.01K tweets illustrated emotional shock. A total of 8.45K tweets depicted the emotion of disgust, which is the least represented.

The emotions of disgust, anger, fear, and negativity contributed to hate content [32–34]. The variation in the expression of hate content is shown in Table 1. The percentage score of all the emotions gradually decreases from April 2021 to June 2021. The gradual decrease in the expression of hatred from April to June depicts the increased involvement of government authorities in social and healthcare development activities.

**Table 1.** Monthly hatred-influencing emotions.

| Month | Disgust (%) | Anger (%) | Fear (%) | Negative (%) |
|---|---|---|---|---|
| April 2021 | 10.24 | 16.8 | 33.27 | 45.5 |
| May 2021 | 9.99 | 16.7 | 32.94 | 43.7 |
| June 2021 | 8.52 | 15.11 | 27.94 | 36.8 |

The most frequent words for emotions in June 2021 are shown in Figure 4. In common with the tweets for May 2021, the emotion of anticipation is depicted through time, public, tomorrow, long, and watch as frequent words. Fight, jab, opposite, court, and battle are frequent words used to show anger. The trust emotion is depicted through dose, import, nation, support, and policy words.

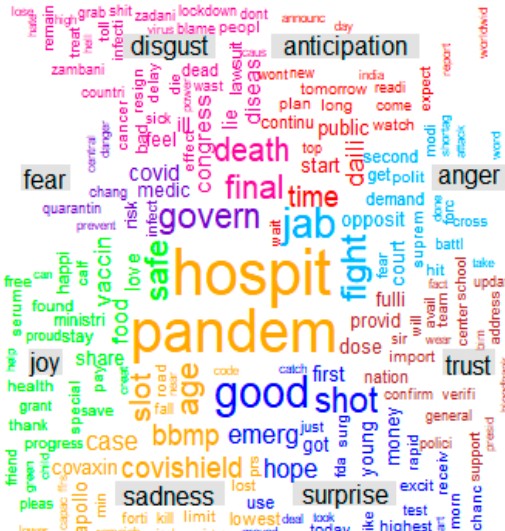

**Figure 4.** Wordcloud of emotions during infectious disease in June 2021.

The frequent words good, hope, shot, young, and money show the emotion of surprise. The emotion of sadness is shown through the words pandemic, case, hospital, covishield, and covaxine. Safe, vaccine, food, share and save words are used to exhibit joy. Fear is shown through governance, quarantine, infect and risk. Disgust is depicted through death, disease, final, lie, and other frequent negative words.

The adjacency matrix of frequent words and their association with collected Twitter content are provided in Table 2. The principal diagonal elements show the frequency of occurrence of words and other elements show the co-occurrence frequency. The association

matrix depicts the fact that the term "COVID-19" is associated more with the words "cases", "vaccine", and "new". Similarly, the word "people" is more associated with the words "COVID-19" and "vaccine".

**Table 2.** The adjacency matrix of frequent terms.

| Terms | COVID-19 | Covid | Vaccine | Cases | People | India | New |
|---|---|---|---|---|---|---|---|
| COVID-19 | 228,606 | 8672 | 22,464 | 24,217 | 9505 | 13,032 | 18,571 |
| Covid | 8672 | 26,416 | 3599 | 2084 | 1199 | 2178 | 1375 |
| Vaccine | 22,464 | 3599 | 40,665 | 358 | 1978 | 1722 | 1050 |
| Cases | 24,217 | 2084 | 358 | 31,335 | 789 | 5034 | 13,267 |
| People | 9505 | 1199 | 1978 | 789 | 22,702 | 984 | 693 |
| India | 13,032 | 2178 | 1722 | 5034 | 984 | 23,556 | 3341 |
| New | 18,571 | 1375 | 1050 | 13,267 | 693 | 3341 | 28,056 |

The content of the adjacency matrix indicates that the term "COVID-19" is central for e-participation and discussion as its association is more with the frequent words than the other terms.

## 5. Discussion

The order of expressing emotions in both the months of April and May 2021 is the same. This indicates that the pattern of expression and showing an awareness and capability level in social media usage for sharing information on infections is similar in April and May 2021. There are more positive tweets than negative tweets, as shown in Figure 1a,b. Twitter content for June 2021 showed the changes in the tweeting pattern of social media users. The high score for the emotion of trust indicates that people are confident about their social media participation and believe in the system for infection regulation.

The next high-scoring emotions, fear, anticipation, and sadness from Figure 1a,b are associated with a negative connotation and try to avoid bad experiences by moving away from the stimulus. The higher score for trust n and the presence of other positivity-associated emotions indicate that people are conscious and competent enough to participate electronically in sharing information over social media. On the other hand, the scores of negative-connotated emotions such as fear, anticipation, and sadness depict the existence of some obstacles to e-participation during the infectious disease.

The changing pattern of expressing emotions in terms of frequent words for e-participation is shown in Table 3. The expression of most of the emotions in each month of the analysis is different. There were 17.73K tweets about the emotion of joy. Some 14.99K tweets contained the emotion of anger. A total of 11.01K tweets illustrated the emotion of shock. Finally, 8.45K tweets depicted the emotion of disgust, which is the least represented. Concerning emotion of trust in April 2021, people started sharing information such as vaccination, the use of masks, etc. With the evolving pattern through May 2021, people discussed the support and requirements of policies to regulate the pandemic situation in June 2021.

The metaphorical expression about an incident involves the target and source entities. The target is represented in terms of its source. Consider the tweet "It is a common enemy and we have put down measures to contain the spread. We Tanzanians are not an isla . . . ", in this tweet, the target infection is stated metaphorically by making an analogy with the enemy. Similarly, the tweets "We are fighting a battle against the common enemy. The way we always keep our business books safe and regu . . . " and "India is in a state of biological war against an invisible enemy (coronavirus), which has imposed unprecedented hea . . . " make an analogy between the infection and the enemy. Metaphorical thinking and talking are possible only when an individual has an awareness and confidence about the topic of discussion. The detailed observation of some random tweets revealed that most of the tweets were expressed metaphorically during the pandemic. This indicates that people were conscious and confident about e-participation and discussion during the pandemic.

**Table 3.** Emotions and awareness of e-participation during infectious disease.

| Emotion | Frequent Words in April 2021 | Frequent Words in May 2021 | Frequent Words in June 2021 | Remarks |
|---|---|---|---|---|
| Anticipation | time, public, tomorrow, long, and watch | time, public, tomorrow, long, and watch | time, public, tomorrow, long, and watch | There are no changes in the expression of anticipation emotion in all three months |
| Anger | fight, strike, destruct, court, and demolish | fight, jab, hit, court, and battle | fight, jab, opposite, court, and battle | The pattern of anger expression is different in each month with a varying set of frequent words |
| Trust | vaccine, mask, wear, nation, and team | mask, import, oxygen, wear, and team | dose, import, nation, support, and policy | Concerning trust the awareness of users gradually changing from curing to prevention |
| Surprise | good, highest, hope, urgent, and case | good, hope, shot, urgent, and highest | good, hope, shot, young, and money | The incidents of surprise have been changed towards the youth and money involvement |
| Sadness | pandemic, covid, hospital, blood, and plasma | pandemic, case, hospital, blood, and plasma | pandemic, case, hospital, covishield, and covaxine | Gradually, people started thinking about the side effects of covid vaccination |
| Joy | safe, stay, happiness, share and save | safe, stay, happiness, share and save | safe, vaccine, food, share and save | Users expressed joy from being safe to get vaccinated. |
| Fear | governance, risk, spike, infect and spread | governance, covid, quarantine, infect and spread | governance, covid, quarantine, infect and risk | The fear factor remains similar in all three months with the least variation. |
| Disgust | death, disease, sick, bad and positive | death, disease, fungus, suffer, and bad | death, disease, final, lie, and bad | People became disgusted with the occurrence of the pandemic and its variants such as fungus |

A tweet " . . . lockdown has a huge effect on our health and emotional being. Here are some simple tricks that will help y . . . " shows the awareness of the tweeter about the health impacts of infection on individuals. Several social media users expressed disgust during the infection. For example, the tweet "Disgusting!! - Medical Staff should be increased immediately so that the existing patients don't succumb to death . . . " describes the healthcare system's condition. This tweet indicates that there is a shortage of healthcare workers and a need to increase the number of healthcare workers to avoid the death of patients. On the other hand, some people expressed their views about trust and confidence. The tweets "Remember everyone. Get the vaccine so you don't get #COVID-19. Trust the science. Do what Government tells you" and "Let's continue to build public trust, counter disinformation, & amp; bring awareness to the importance of #COVID-19 vacci . . . " are two examples of trust expressed by social media users.

Several people supported persons with disabilities through their expression using social media. A social media user tried to help persons with disabilities by tweeting "Join us this Saturday as we discuss the current pandemic and its effect on disability. Details are on the flyer", and organizing a discussion. The tweeter intended to alert persons with disabilities to avoid the impacts of infection. Another Twitter user tweeted "Govt plans to start mobile van vaccination facility for people who are bedridden and people with disability after . . . " in support of persons with disabilities. This is a motivational tweet and shows that persons with disabilities are not excluded from the system; rather, they are considered for priority care. Another person tweeted saying "Why some #disabled people need to be further up the queue for the #COVID-19 jab". This social media user is conveying a message to vaccination centres about serving persons with disabilities as a priority.

The tweet "Our disability-inclusive #COVID-19 response and recovery aims to ensure accessible public health information, imple . . . " indicates the dedication of the healthcare system to persons with disabilities. Another tweet " . . . clinics can be an overwhelming space for those with an intellectual disability. Mock clinics can help thos . . . " indicates the requirement of specific healthcare centres for persons with disabilities. The content of

these sample tweets indicates that the social media users are e-participating consciously and competently to express their support for persons with disabilities.

## 6. Conclusions

The Twitter social media data were collected and analyzed for the capability and awareness of users in terms of emotions. As the positivity-associated emotions hoped for good incidents to take place, trust and other positive emotions showed the awareness and capability of people to participate electronically in the exchange of information online. Negative emotions such as fear, anticipation, and sadness may have acted as an obstacle to e-participation during the infection. The involvement of people in deliberations concerning the infection for supporting persons with disabilities increased their awareness and competence in e-participation. There exists an evolving pattern of expression showing the intent of curing the infections of the pandemic to prevent their spread. As the governing authorities adopted some strategies to control the pandemic, there was a reduction in hatred-containing tweets from April to June 2021.

The results showed that fatigue- and fear-mongering metaphors were prominent during the infection deliberations. The usage of war- and enemy-related metaphors was prevalent. The content of some randomly selected tweets shows that people expressed their concern in regard to persons with disabilities. A lack of education and inadequate capability for complex experience limited impactful e-participation. Further, the development of false awareness among e-participants led to a problematic environment.

Capability development in a catastrophic environment requires a conscious effort. Problem-specific disability responsiveness during the infection was essential to bring about meaningful e-participation. The diffused generic information on the catastrophe did not necessarily produce a conscious experience of disability. Existing e-participation studies assessed the capability less from the conscious levels. On the whole, metaphors were not utilized for awareness assessment due to a lack of automated tools and region specificity. Inclusive e-participation requires integrating experience with awareness developed over the available information. The current study depicts the capability-awareness framework for inclusive e-participation.

The present work used only Twitter social media data, equidistant samples, and topic modelling concepts to identify the capability, consciousness, and tweeting patterns of social media users. In the future, the work can be extended to multiple social media content and non-equidistant samples by applying machine-learning techniques.

**Author Contributions:** Conceptualization, S.A.; methodology, S.A., N.C. and V.K.; software, N.C.; validation, S.A., S.D. and R.R.P.; formal analysis, S.A., V.K.; investigation, S.A. and N.C.; resources, S.A.; data curation, S.A. and N.C.; writing—original draft preparation, S.A. and N.C.; writing—review and editing, S.A. and S.D.; visualization, S.A. and V.K.; supervision, S.A.; project administration, S.A.; funding acquisition, S.A. All authors have read and agreed to the published version of the manuscript.

**Funding:** This paper is largely an outcome of the Research project sponsored by the Indian Council of Social Science Research (ICSSR). However, the responsibility for the facts stated, opinions expressed, and the conclusions drawn is entirely that of the author.

**Data Availability Statement:** Not Applicable; the study does not report any data.

**Acknowledgments:** The scholar S.A. is the awardee of ICSSR Special Call for Studies Focusing on Social Science Dimensions of COVID-19 Coronavirus Pandemic.

**Conflicts of Interest:** The authors declare no conflict of interest.

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
