# Peer review of "Hate and False Metaphors: Implications to Emerging E-Participation Environment"

_futureinternet, doi:10.3390/fi14110314_

Round 1

Reviewer 1 Report

The article meets the requirements for scientific articles and offers an important contribution in the area. The subject is current and interesting, and it shows the importance of the problem. The structure is logical and easy to follow. The results are useful. In my opinion this is a well-written and valuable paper. I like the article very much. I only suggest adding in the Conclusions section further research directions and limitations.

Author Response

kindly check the attached file

Reviewer 2 Report

This proposed paper has a popular main title, it is about the use of ‘hate’ and ‘false’ while communicating over social networks. The text starts immediately with dialectics in corresponding wording. This does not make the text easy to understand for the average audience. The authors see the journal as attractive for only a small number of experts. For new branches of science, the key for wrestling an independent existence is the readability for colleagues related corners.

The manuscript is well structured by copying from well-received publications. It copies the universally valued pillars, which consequently leaves holes in specific cases. In other words, the text presents the shallow situation that shows the applicability of Natural Language Processing (NLP) to also the Fake Truth Detection. This does not surprise the reader by itself. However, the reader will hunger for more. The section “Results” is a mere listing of the Frequency Analysis, the results are noted in the section “Discussion”. This is together a very shallow data processing.

The analysis is a nice graphical presentation of the linear multi-node tree. These art-like figures have no link to the aim of the research. Steered by the paper title, one expects at least the growth of an E-participation environment in terms of hate and false metaphors. This is also not made apparent in a paragraph on (sub)research goals at the end of the introduction, nor in a validated part of the Conclusion that makes the achievements of the research. Consequently, the paper is readable and interesting but remains aimless.

A linear tree carries the word discrimination. This tends to clogging soon if the same word distance is used on every tree level. Non-linear discrimination is often a minimum requirement. Assuming that the database is pre-labelled, the critical events where words are assigned to the wrong label set can be determined. In the time-series provided by a Twitter dataset gives additional help for correct discrimination and identify critical events.

For this reason, I would appreciate to combine Figs, 1, 3 and 5 to see the emotions in a moving window. On the other hand, equidistant sampling (such as 1 month apart) is not a reliable way to find trends.

The paper is showing the real meaning at the end of the Discussion. In a Metaphor by itself, it suggests a direct line between NLP and fake/hate disguise to detect emotion manipulation in social networks. The question is whether this can be ever effective. This is not a matter of a simple NLP application. The author gives a number of examples for the basic functional requirements. This is at the moment already an achievement but we have to keep an open eye for the long-term requirement. It is advised here to add a number of such functional needs for the future at the end of the conclusions section.

Author Response

kindly check the attached file

Reviewer 3 Report

Dear Authors,

The paper is interesting. However, I'm a big fan of statistical analysis and data science, but your paper is acceptable. Please improve conclusion part. It means that, please add some suggestion and limitations. Meanwhile, please add some new and more relevant references.

Best regards,

One of reviewers.

Author Response

kindly check the attached file

Round 2

Reviewer 2 Report

When asked whether the comments in a review are adequately handled, you need instruction on how to measure this. When the reaction is given just a short time, there is little to expect. There is no time given to consider the comments given by the reviewer. That is unfair for the authors and to the reviewers as well. So the question remains whether there is some truth in the paper and whether the readers could actually learn from it.

The authors have made some effort to handle the review comments, but some remarks go beyond their recognition. So Figure 1 is a clear change in comparison to the single figures earlier, but hardly an improvement. It is still not easy to interpret. The parts are simply brought closer but not integrated on normalized axis’. I appreciate that the authors tried their best, but that is about it.

The usability of the paper is increased by adding some hints. Still it does not provide a thread of new wisdom. At best it can picture some data, but generalization is still far away. There is a long history of data clustering experimentations with humorous results. The technology used for the introduced distinction between “fake and fate” still leaves a lot to be desired.

But, if the purpose of the publication is to add to the public discussion, the text may reach the goal. With that in mind, I support the proposal to have this text published. Let’s see what happens and along the line learn where the real value is as perceived by the readers.
